# Identification and Characterization Roles of Phytoene Synthase (*PSY*) Genes in Watermelon Development

**DOI:** 10.3390/genes13071189

**Published:** 2022-07-01

**Authors:** Xufeng Fang, Peng Gao, Feishi Luan, Shi Liu

**Affiliations:** 1College of Horticulture and Landscape Architecture, Northeast Agricultural University, Harbin 150030, China; 18800469189@163.com (X.F.); gaopeng_neau@163.com (P.G.); 2Key Laboratory of Biology and Genetic Improvement of Horticulture Crops (Northeast Region), Ministry of Agriculture and Rural Affairs, Harbin 150030, China

**Keywords:** flesh color, phytoene synthase, gene expression, functional analysis, carotenoid biosynthesis pathway

## Abstract

Phytoene synthase (*PSY*) plays an essential role in carotenoid biosynthesis. In this study, three *ClPSY* genes were identified through the watermelon genome, and their full-length cDNA sequences were cloned. The deduced proteins of the three *ClPSY* genes were ranged from 355 to 421 amino acid residues. Phylogenetic analysis suggested that the *ClPSYs* are highly conserved with bottle gourd compared to other cucurbit crops PSY proteins. Variation in *ClPSY1* expression in watermelon with different flesh colors was observed; *ClPSY1* was most highly expressed in fruit flesh and associated with the flesh color formation. *Cl**PSY1* expression was much lower in the white-fleshed variety than the colored fruits. Gene expression analysis of *ClPSY* genes in root, stem, leaf, flower, ovary and flesh of watermelon plants showed that the levels of *ClPSY2* transcripts found in leaves was higher than other tissues; *ClPSY3* was dominantly expressed in roots. Functional complementation assays of the three *ClPSY* genes suggested that all of them could encode functional enzymes to synthesize the phytoene from Geranylgeranyl Pyrophosphate (GGPP). Some of the homologous genes clustered together in the phylogenetic tree and located in the synteny chromosome region seemed to have similar expression profiles among different cucurbit crops. The findings provide a foundation for watermelon flesh color breeding with regard to carotenoid synthesis and also provide an insight for the further research of watermelon flesh color formation.

## 1. Introduction

Approximately 750 carotenoids have been described in nature, and that number continues to grow [1]. Plants attract insects and birds for pollination and seed dispersal, in large part because of carotenoids, which confer different colors to plant parts. In addition, as part of the human diet, carotenoids are the precursor of vitamin A and the basis of many nutrients; carotenoids also have antioxidant activity. Although these compounds are essential to health, humans cannot synthesize them [2]. The first step is mediated by nuclear-encoded PSY, which catalyzes the conversion of two GGPP molecules to phytoene in plants [3]. 15-*cis*-phytoene is the first colorless carotenoid produced by PSY, a rate-limiting enzyme in the carotenoid biosynthesis pathway [4,5]. The decrease in total carotene content in *Arabidopsis* mutant etiolated seedlings is consistent with the requirement of galactose for PSY activity [6].

Many plant species contain three to five *PSY* family members, while in *Arabidopsis,* there is only one. Three *PSY* genes (*CpPSYA*, *CpPSYB* and *CpPSYC*) have been identified in *Cucurbita pepo*, though only the abundance of the *CpPSYA* transcript affects the carotenoid biosynthetic pathway [7]. Apple has six *MdPSY*s, however, which mainly involve *MdPSY1* and *MdPSY2* with relatively high enzymatic activity and expression levels [8]. The function of *PSY* was affected by many regulatory patterns. Different PSY members also have the tissue-specific expression characteristics. In tomato [9] and citrus [10], *PSY1* was specifically expressed in fruits, while *PSY2* and *PSY3* were mainly detected in leaves and roots. The silencing of *SlPSY1* resulted in a yellow fruit color from red [11]. The fruit phenotype did not overtly change when *SlPSY2* and *SlPSY3* were silenced, suggesting that *SlPSY1* plays the most significant role in tomato fruit. *SlPSY2* functions mainly in chloroplast-containing tissues [9]. *PSY-A* was found in watermelon’s whole tissue (including fruits), and *PSY-B* was found in watermelon leaves and roots [12]. The transcriptome dynamics at certain watermelon fruit developmental stages suggested that lycopene accumulation affects the up-regulation of *ClPSY* [13]. Different *PSY* members were also expressed in unique locations within the chloroplast, such as thylakoid membranes associated with plastoglobules or the envelope membrane and stroma [14]. In maize, *ZmPSY1* is involved in the formation of distorted plastid shape and fiber phenotype [15]. Furthermore, variant localization may also alter enzyme activity. When *PSY* was overexpressed in *Arabidopsis*, *PSY* enzyme activity was detected only in the membrane-bound form but not in the matrix localized form [16]. Transcription factor was another *PSY* regulator for carotenoids accumulation. In tomato, the ripening-inhibitor (*RIN*) has been shown to regulate carotenoid concentration in fruits by interacting with the *SlPSY1* promoter [17]. *SlBBX20* in tomato can activate the expression of *SlPSY1* by directly binding to the G-box motif of its promoter, resulting in increased expression of *SlPSY1* for carotenoid accumulation, presenting dark green color in fruits and leaves [18]. Transcriptomic data showed that the expression trend of *AdMYB7* in *N*. *benthamiana* was consistent with that of *NdPSY* [19]. Overexpression of *CsMADS6* in citrus callus can directly combine with *CsPSY* promoters to increase the carotenoid content [20]. *PSY* expression is also regulated by the feedback of carotenoid pathway products [21]. The overexpression of *AtCYP97A* in carrots increased α-carotene accumulated to lutein but decreased PSY protein level and led to the decrease in carotenoids [22]. *PSY* can affect carotenoid accumulation at the protein level post-transcriptionally. PSY protein levels were negatively regulated by carotenoid metabolites and total carotenoid content. The expression level of *PSY* is regulated by OR/OR-like proteins in *Arabidopsis* and potato [23,24]. As a post-transcriptional regulator of *PSY*, *PIF1* is involved in the development of chlorophyll and chloroplast and the production of carotene [25]. The orange flesh color (β-carotene accumulated) in melon was affected by the *CmOr* (encoding a plastid-targeted protein) gene [26]. Further research confirmed that the *CmOr* had little effect on the *CmPSY1* expression level but highly affected CmPSY1 protein levels [27]. The decrease in Clp activity in clpc1 leads to the decrease in PSY protein transformation. The OR protein enhanced the stability of the PSY protein and increased the activity of PSY in clpc1 [28].

To date, *PSYs* have been studied extensively in other crops but less so in cucurbitaceae, especially watermelon. In this study, we cloned three *ClPSY* members from the watermelon genome, characterized and compared the complete protein coding sequences (CDS), analyzed the expression profiles in different tissues and evaluated the expression patterns during fruit development in fruits of different colors. In addition, the *ClPSY* gene family was investigated with regard to enzyme activity using a heterologous complementation system. The expression pattern and genomic synteny of different *PSY*s among the cucurbit crops also exhibited that some *PSY*s in the genomic synteny may have a similar gene function.

## 2. Materials and Methods

### 2.1. Plant Materials

Six watermelon cultivars with different flesh colors: white-fleshed ‘ZXG0000’ (PI 494532); orange-fleshed ‘ZXG0077’ (Charleston); pink-fleshed ‘ZXG1594’ (Javrijsky); red-fleshed ‘ZXG55’ (Lanzhou Huapi); ‘ZXG1549’ (Qing Pi); and ‘ZXG0079’ (Huangbao No.1) were used in this study. All the plants were planted in the greenhouse at the Xiang Yang Agricultural Experiment Station of Northeast Agricultural University, Harbin (44°04′ N, 125°42′ E), China, in 2018. Tissue samples were collected at 10, 18, 26, 34 and 42 days after pollination. Three watermelon fruits with similar development status and no mechanical damage were collected at each development stage of each variety and stored at −80 °C for subsequent experiments, including DNA and RNA extraction. ‘ZXG0000′ belonged to the *C*. ssp. *mucosospermus* with a late fruit setting, so the fruit samples were collected from this line at 10, 18, 26, 34, 42 and 50 DAP. Roots, stems, flowers, ovaries and leaves were collected from ‘ZXG0079’.

### 2.2. RNA Isolation and cDNA Synthesis

Total RNA was performed according to the instructions of Novogene (Beijing) RNA extraction kit, which was extracted from all tissues, including roots, stems, leaves, flowers and ovaries, at each fruit stage. The total RNA was detected by 1% (*w/v*) agarose gel electrophoresis. In addition, the non-degraded RNA was selected as OD260: OD280 > 1.80 for cDNA synthesis. cDNA synthesis was performed according to the qPCR RT Kit (Code No. Fsq-101, TOYOBO, Tokyo, Japan). The first step was to denaturate RNA, that is, the qualified RNA was placed in a 65 °C metal bath for 5 min and then immediately put in an ice box for cooling, so as to improve the reverse transcription efficiency. The cDNA after reverse transcription was stored at −20 °C for qRT-PCR and coding region cloning.

### 2.3. Identification and Cloning of ClPSY Genes

To identify members of the *ClPSY* gene family, the Cucurbit Genomics Database (https://www.cucurbitgenomics.org/watermelon97103 genome v2/; accessed on 15 January 2022) was searched for putative *ClPSY* genes (Table 1) from watermelon genome version of 97103 v2. To amplify full-length cDNA, specific primers were designed according to the sequences of putative *ClPSY* genes using Premier 6.0 software. The primer sequences, amplicon sizes and accession numbers are shown in Table 2. Polymerase chain reaction (PCR) was performed using cDNA from flesh tissues at 34 days after pollination (DAP) as the template with the reported conditions [29]. Target genes were purified using a gel extraction kit (Kangwei, Beijing, China). pMD18-T vector was used for cloning vector linkage (TaKaRa, Tokyo, Japan) and introduced into *Escherichia coli* strain DH5α.

### 2.4. Expression Pattern Analysis of ClPSYs

Specific primers amplified to approximately 150 bp were designed using Premier 6.0 software. In addition, *ClYLS8* (*Cla020175*) was used as the reference in quantitative real-time PCR [30]. The primer sequences, amplicon sizes and accession numbers are shown in Table 2. The transcript levels of genes were evaluated by qPCR using the QTOWER (Analytik Jena, Germany) and SYBR Green Master Mix (Novogene, Beijing, China). Each qPCR had three technical repeats. The PCR template with the reported conditions [29]. SPSS v21.0 software (Chicago, IL, USA) was used for real-time quantitative PCR data, and 2^−^^△△CT^ method was used to calculate the relative expression, finally plotted using the Prism 7.0 software [31].

### 2.5. ClPSY Functional Complementation in E. coli

A heterologous complementation assay was performed to assess the functions of all three ClPSY proteins. Two *E. coli* BL21-Gold strains were used in the experiment. Cells harboring the pACCRT-EB plasmid (EB) that carries the bacterial carotenogenic genes *crtE* and *crtB* were used as a positive control; these cells accumulate phytoene. Cells harboring the pACCRT-E plasmid (E) that carries the bacterial gene *crtE* produce the GGPP enzyme in bacteria, which is required for *ClPSY* function to catalyze the conversion of two molecules of GGPP into phytoene. The *Cl**PSY* coding sequences were amplified and cloned into pET to generate pET-*Cl**PSY1*, pET-*Cl**PSY2* and pET-*Cl**PSY3* vectors. Each pET-*ClPSY* expression vector was transformed into strain E to test for enzyme activity. An empty pET vector was also transformed as a negative control. Positive clones were grown in 15 mL Luria–Bertani (LB) liquid medium containing appropriate antibiotics (100 mg/L ampicillin, 34 mg/L chloramphenicol) overnight at 37 °C. A 1 mL aliquot of culture was inoculated into 100 mL LB liquid medium and grown at 37 °C until reaching an optical density of 0.5 to 0.8 at 600 nm. The cells were then grown at 30 °C for 48 h in the dark to maximize carotenoid production. All experiments were performed in triplicate. For extraction, cultures were centrifuged for 20 min in the dark at 4000× *g* and washed with distilled water. The pellets were re-suspended with 5 mL of methanol containing 1% butylated hydroxytoluene (BHT), and the cells were sonicated twice with 1 min pulses on ice. The homogenate was centrifuged at 4000× *g* for 15 min, and the extracts were dried using a MICRO-CENVAC (NB-503CIR, N-BIOTEK, Bucheon, Korea) for 2 h at 45 °C following Ampomah-Dwamena et al. [8].

The carotenoid extracts were suspended in 1 mL acetone-hexane (V:V = 1:1) and then filtered through a 0.45 µm syringe filter into a high-performance liquid chromatography (HPLC) vial. Carotenoid analysis was performed using an HPLC instrument (Waters, Milford, MA, USA) equipped with a binary HPLC pump (1525, Waters), an auto-sampler (2707, Waters) and a photodiode array detector (2998, Waters) with a 4.6 × 250 mm, 5 mm column (LC ZORBAX SB-C18; Agilent Technologies, Palo Alto, CA, USA) (modified from Wang et al. [13]). The column temperature was 25 °C, and the column flow rate was 1.00 mL/min. The elution of phytoene was observed at 286 nm, and the retention time was the minimum, according to the standard.

### 2.6. Phylogenetic Tree Construction and Genome Collinearity Analysis among Cucurbit Crops

We also identified *PSY* genes in other cucurbit crops from the Cucurbit Genomics Database (https://www.cucurbitgenomics.org/; accessed on 15 January 2022) with the following crops and genome versions: *Cucumis melo* (DHL92, v3.6.1 [32]), *Cucumis sativus*, (Chinese Long, v3 [33]), *Cucumis sativus*, (wild cucumber, PI 183967 [34]), *Cucurbita maxima* (*Rimu* [35]), *Cucurbita moschata* (*Rifu* [35]), *Cucurbita pepo* (*Zucchini*, BGV004370 [36]), *Lagenaria siceraria*, (bottle gourd, USVL1VR-Ls [37]) and *Cucurbita argyrosperma*, (silver-seed gourd [38]). The genes orthologous to the three *ClPSYs* were chosen as the homologous genes. The amino acid sequences for each *PSY* were extracted and aligned for trimming by using the trimAI software [39]. The structure of *ClPSY* genes and multiple sequence alignments were analyzed using the Bioedit 7.0 software. Phylogenetic relationships were inferred using the UPGMA method, and a dendrogram was constructed with the MEGA-X program [40]. The phylogenetic tree and the homologous gene structures were colored and drawn with Evolview (https://www.evolgenius.info/evolview/#login/; accessed on 15 January 2022). The collinearity analyses of cucurbit crops genome were analyzed and plotted with TBtools (v 1.046).

### 2.7. Public RNA-Seq Data Analysis

The published RNA-seq data of watermelon (PRJNA221197, PRJNA270773, PRJNA338036 and SRP012849), melon (PRJNA286120, PRJNA288543, PRJNA314069 and PRJNA383830), cucumber (PRJNA312872), *Cucurbita maxima* (*Rimu*) and *Cucurbita moschata* (*Rifu*) (PRJNA385310), *Cucurbita pepo* (*Zucchini*) (PRJNA339848) and bottle gourd (PRJNA387615) were used for the *PSY*s expression pattern analysis. The means for each read per kilobase per million mapped reads (RPKM) value were retrieved from the above-mentioned respective BioProjects and plotted with the Prism 7.0 software.

## 3. Results

### 3.1. Sequence Analysis of ClPSY Genes and Genes Expression during Watermelon Flesh Development

Three *PSY* genes were identified (*ClPSY1*, *ClPSY2* and *ClPSY3*, Table 1) through the watermelon draft genome from the Cucurbit Genomics Database. We further cloned, characterized and compared the CDS of three *ClPSY**s* and then analyzed the nucleotide and amino acid sequences. As illustrated in Figure 1, *ClPSY1*, *ClPSY2* and *ClPSY3* encode polypeptides consisting of 421, 387 and 355 amino acids, respectively, and contain characteristic motifs, including a putative phytoene synthase active site (DXXXD), which is crucial for *PSY* activity [41]. Protein sequence identity among the three *ClPSY* homologs ranged from 68 to 75%. At the amino acid level, *ClPSY1* showed 75% identity with *ClPSY2*, *ClPSY2* showed 72% identity with *ClPSY3*, and *ClPSY1* showed approximately 68% identity with *ClPSY3*.

The flesh development and color formation among different watermelon accessions are exhibited in Figure 2. During the early stage (0 to 18 DAP), the six watermelon accessions remain white fleshed; the carotenoids obviously start to generate at 26 DAP, except for the white flesh color accession. From 26 DAP to 42 DAP, the pigments seem to increase constantly until the maturation period. ‘ZXG0000’ (white fleshed) did not change through all the development stages. To explore the relationship between *ClPSY* genes and flesh color development, the transcript level was analyzed in five or six developmental stages for the six watermelon accessions. For all the three *ClPSY* genes, *ClPSY1* transcripts were much more abundant than *ClPSY2* and *ClPSY3* transcripts in all varieties through the entire fruit development period (Figure 3a–c). In the first three or four stages, the expression of *ClPSY1* was increased continuously and declined in the last stage (50 DAP for ‘ZXG0000’ and 42 DAP for the other five materials) for all the experimental watermelon accessions. The expression of *ClPSY1* reached the highest level at 34 DAP for all varieties, except for the white flesh accession ‘ZXG0000’ (reached maximum at 42 DAP) (Figure 3a). *ClPSY1* transcripts were highly expressed in red-fleshed ‘ZXG0079’, ‘ZXG1549’ and ‘ZXG0055’, more so than in the pink flesh accession ‘ZXG1594’ when the fruit began to mature and the flesh accumulated the most pigment, implying that the period from 26 to 34 DAP was an important stage for the color formation. Even in the same flesh color group (red group: ‘ZXG0079’, ‘ZXG1549’ and ‘ZXG0055’), the expression levels of *ClPSY1* were still different, perhaps due to the accumulation amount of total carotenoid in each stage for different accessions. Interestingly, *ClPSY1* transcript levels were a little higher in orange ‘ZXG0077’ than in the red and pink materials, indicating that ‘ZXG0077’ may accumulate more total carotenoids than the other accessions.

For *ClPSY2* and *ClPSY3*, no sensible expression pattern and variation could be detected in all the six accessions, both among different flesh colors and development stages, implying that this gene may not contribute mainly to the flesh color formation (Figure 3b,c). In *ClPSY2*, the expression pattern could be divided into three groups: two red accessions (‘ZXG1549’ and ‘ZXG0079’) were similar, while ‘ZXG1549’ and ‘ZXG0079’ were expressed more substantially than ‘ZXG0055’ and ‘ZXG1594’. The colorless accession ‘ZXG0000’ performed a medium gene expression, based on our results (Figure 3b).

The expression patterns of three *ClPSY*s were further analyzed for different organs (leaf, flower, ovary, root and stem) with red-fleshed ‘ZXG0079’ at the mature stage (42 DAP, Figure 3d). All three *ClPSY*s could express in each organ with different levels. Except in the flesh, the *ClPSY1* transcript performed a moderate expressed variation in leaf, flower, ovary and stem, except for the root, with the lowest expression. The transcription patterns of *ClPSY1* and *ClPSY2* were similar in the stem and ovary but performed significantly differently in the leaf and flower. In particular, the expression of *ClPSY2* was markedly higher in the leaf than the ovary and flower, implying that *ClPSY2* may be predominantly expressed in the leaf. *Cl**PSY3* may specifically express in roots due to its considerable expression amount and the obviously lower expression of other *ClPSY*s.

### 3.2. Functional Analyses of ClPSY Proteins in Watermelon

Standard bacterial complementation method was applied to evaluate catalytic activities of three watermelon *ClPSY* genes-encoding enzymes. Strain EB, which produced phytoene, was used as a positive control; strain E produced GGPP in bacteria requiring functional *ClPSYs* to produce phytoene. Plasmids carrying cDNA fragments of the *ClPSY*s were transformed into strain E. Based on the results of the HPLC analysis, we found peaks with retention time (from 11 to 12 min) and spectral qualities that were the same as the positive control in extracts of cells expressing each *ClPSY* gene (Figure 4), suggesting that all the *ClPSY*s encoded enzymes catalyzed the conversion of GGPP to phytoene in the cells. These results indicate that these *ClPSY*s encoded functional enzymes under these heterologous complementation system conditions.

### 3.3. Synteny Analysis of PSY Members among Different Cucurbit Crops and Their Expression Patterns with the RNA-Seq Data

Three homologous *PSYs* were detected in melon, cucumber, wild cucumber and bottle gourd, while for the *Cucurbita* crops, there were five homologous *PSYs* (Table 3). Compared with the other cucurbit crops, we detected two *PSY1* and *PSY3* homologous genes through the whole genome only in the three *Cucurbita* crops. The phylogenetic results indicated three major clades through all the *PSYs* (Figure 5). For all the cucurbit crops, in each phylogenetic clade, the cucumber (including wild cucumber PI 183967) and melon clustered together, while the watermelon was close to the bottle gourd. *PSYs* in *Cucurbita* members were clustered into one sub-branch for each *PSY* member, and the different copies were also clustered separately for homologous *PSY1* and *PSY3*. We further checked the syntenies of the *PSY* homologous genes among the cucurbit crops. All the *PSY* homologous genes were located in the synteny chromosome region (Figure 6).

In order to check the *PSYs* expression pattern in different cucurbit crops, we used the published RNA-seq data to extend the expressed tendency. Some of the homologous genes clustered together in the phylogenetic tree and located in the synteny chromosome region seemed to have similar expression profiles. For *ClPSY1*, which we found expressed specifically in the watermelon flesh (Figure 7a,b), the homologous genes in melon (*MELO3C025102.2*, *CmPSY1*, Figure 7c,d), cucumber (*CsaV3_5G020340*, *CsPSY1*, Figure 7e), *Cucurbita moschata* var. *Rifu* (*CmoCh04G023720*, *CmoPSY1-1*, Figure 7g) and *Cucurbita pepo* subsp. *pepo* (*Cp4.1LG01g19670*, *CpPSY1-1* and *Cp4.1LG13g05570*, *CpPSY1-2*, Figure 7j,k) were also mainly expressed in the flesh and increased as the fruit developed, reaching the peak when fruit was mature or near mature. The expression level of *ClPSY1* in the flesh was obviously elevated compared with the fruit rind (Figure 7b). In melon, the expression of *CmPSY1* was increased with the fruit development until the 30 DAP but did not correspond with the carotenoids accumulation. The orange and green flesh color melon fruits exhibited a similar expression pattern and level. Compared with watermelon, *CmPSY1* was expressed in both fruit and leaf (the expression level in leaf was a little higher than the fruit) at the mature stage (Figure 7d). Nonetheless, the gene expression differences were not significant between the green and orange flesh (Figure 7c). In cucumber, the expression of *CsPSY1* also exhibited an obvious expression in the male flower and unfertilized ovary peel (Figure 7e). In *Cucurbita moschata* var. *Rifu*, only one *CmoPSY1* accumulated in abundance at the transcription level in the fruit, while the other one exhibited a relatively high expression level in the leaf (Figure 7f,g). In *Cucurbita maxima* (*Rimu*), the two *PSY1* homologous genes, *CmaCh04G022670* and *CmaCh15G007680,* showed an abundant expression level in the stem compared to the other tissues. None of the *CmaPSY* members were specifically expressed in the flesh in *Cucurbita maxima* (*Rimu*), based on the published RNA-seq data (Figure 7h,i). In the bottle ground, *Lsi09G008920* (*LsiPSY1*) was highly expressed in the leaf, which may due to the low carotenoid accumulation in its flesh (Figure 7l).

For *ClPSY2* homologous genes: *MELO3C014677.2* (*CmPSY2*, Figure 8c,d), *CmoCh04G000340* (*CmoPSY2*, Figure 8f), *CmaCh04G000310* (*CmaPSY2*, Figure 8i), *Cp4.1LG16g09330* (*CpPSY2*, Figure 8h), *Lsi07G009710* (*LsiPSY2*, Figure 8g), *CsaV3_4G023380* (*CsPSY2*, Figure 8e), the expression pattern was quite different. In melon, *CmPSY2* mainly expressed in the male flower, female flower and root, while it was obviously low in the fruit and leaf. *CmPSY2*, *CpPSY2* and *CsPSY2* were all expressed in the flesh but did not correspond with the fruit development stages. In the bottle gourd, *LsiPSY2* was mainly expressed in the fruit and root. The *ClPSY3* homologous genes in melon (Figure 9c,d), cucumber (Figure 9e), Cucurbita maxima (*Rimu*) (Figure 9i), Cucurbita moschata var. Rifu (Figure 9f,g) and bottle gourd (Figure 9l,m) were expressed mainly in the leaf. *CsPSY3* was also mainly expressed in the female and male flower. In the flesh, *CmPSY3* also gradually declined from 10 DAP to maturation. *CpPSY3-1* and *CpPSY3-2* could also be detected in the flesh with a low expression level but did not correspond with the fruit development (Figure 9j,k).

## 4. Discussion

The accumulation of carotenoids in the watermelon plant and fruit may be a result of the three *ClPSY*s function composition, and *ClPSY1* appears to be most important in the flesh, based on our research result. Members of the *PSY* family might have different effects or might act coordinately in some processes of the carotenoid metabolism regulation during the growth of watermelon and other plants. We also conducted an experiment investigating *ClPSY1* expression at different stages of fruit development in watermelon varieties with different flesh colors, and the results were the same as those obtained in previous works [12,13]. *ClPSY1* transcript levels were very low initially and reached a maximum with the progression of fruit ripening in colored watermelon flesh (Figure 3a). High expression of *ClPSY1* resulted in a large accumulation of phytoene, which provided plenty of primary ingredients for carotenoids synthesis [29]. These demonstrated that *ClPSY1* might perform a crucial function of regulating carotenoid synthesis in the watermelon, consistent with many previous studies. According to the *Cucurbita pepo* genome and RNA-seq data, we identified five *PSY* homologous genes and two homologous (*CpPSY1-1* and *CpPSY1-2*, clustered with *ClPSY1* in one clade) genes exhibited the same trends as the fruit development, which was consistent with the flesh color formation. The RPKM values of the two genes also exhibited an obvious difference between the low and high carotenoids accumulated in the *Cucurbita pepo* accessions.

The analysis of the *ClPSY* gene family expression patterns in different organs was performed to show that the three *ClPSY* genes have specialized roles in watermelon organs. The three members were found to be expressed in all tested tissues but at significantly different levels in different plant organs. Such organ-dependent expression pattern in *PSY* has also been found in other plants. Our data revealed that, in watermelon, *ClPSY2* might regulate the leaf, ovary and flower development, and *ClPSY3* is preferentially expressed in the root. This can also be reported in tomato [9], citrus [10] and cassava [4]. For other cucurbit crops, the expression of *PSY* members could also be detected in different plant organs, but the main tissues were quite different among the crops. In cucurbita crops, two copies of *ClPSY1* and *ClPSY2* homologous genes could be detected from the genome data. The two copies of each member always showed the same expression pattern. Interestingly, in *Cucurbita moschata* var. *Rifu*, the two members of the *ClPSY1* homologous genes exhibited different tissue specificity expression patterns. *CmoCh15G007980* was in the leaf and *CmoCh04G023720* was the in fruit.

Based on the previous publications, the genetic relationships of watermelon and bottle ground were close to each other, and the melon was close to the cucumber [42,43]. In our results, the phylogenetic tree also showed a similar trend, but the gene function did not correspond with the genetic relationship. Genes (or QTLs) located in the collinear chromosome regions always exhibited the same gene function, such as *CmCLV3*, *CsCLV3,* and some QTLs were related with the fruit shape and seed size in cucurbit crops [44,45,46]. In our results, the *PSY* members showed a high collinear relation among all the cucurbit crops, but the function may not have been the same according to the published RNA-seq data. In the high carotenoid accumulation crops, such as watermelon, melon, *Cucurbita pepo* and *Cucurbita maxima* (*Rimu*), the *ClPSY1* showed a similar expression pattern and gene function.

## 5. Conclusions

*ClPSY1* is the key gene controlling carotene accumulation in the flesh, while *ClPSY2* might regulate the leaf, ovary and flower development, and *ClPSY3* is preferentially expressed in the watermelon root. The three *ClPSY* members in the watermelon could encode active enzymes. Some of the homologous genes clustered together in the phylogenetic tree and located in the synteny chromosome region seemed to have similar expression profiles among different cucurbit crops.

## Figures and Tables

**Figure 1 genes-13-01189-f001:**
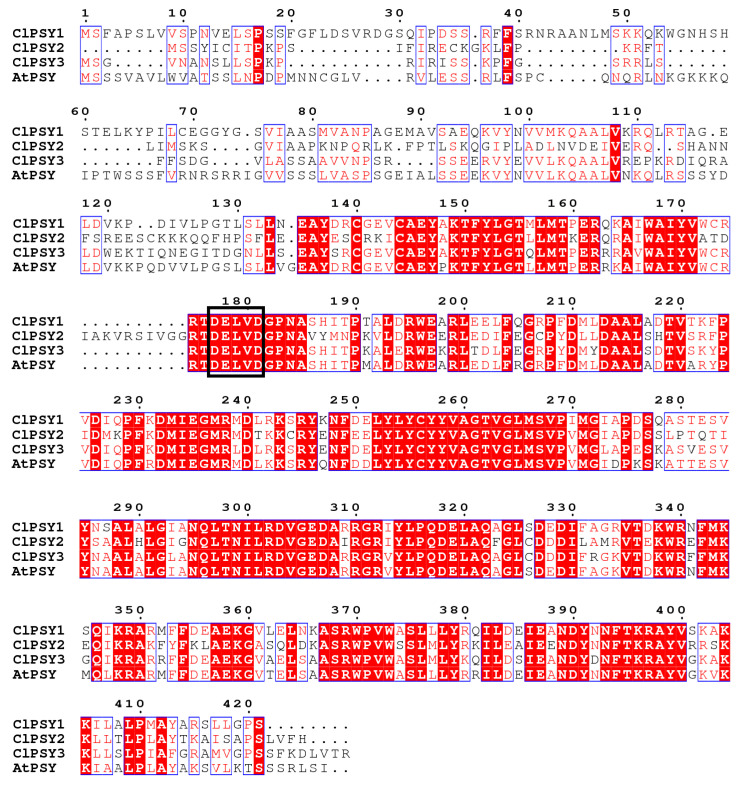
Multiple alignment of watermelon *ClPSY* genes. The hypothetical active site (DXXXD) is shown in the black box.

**Figure 2 genes-13-01189-f002:**
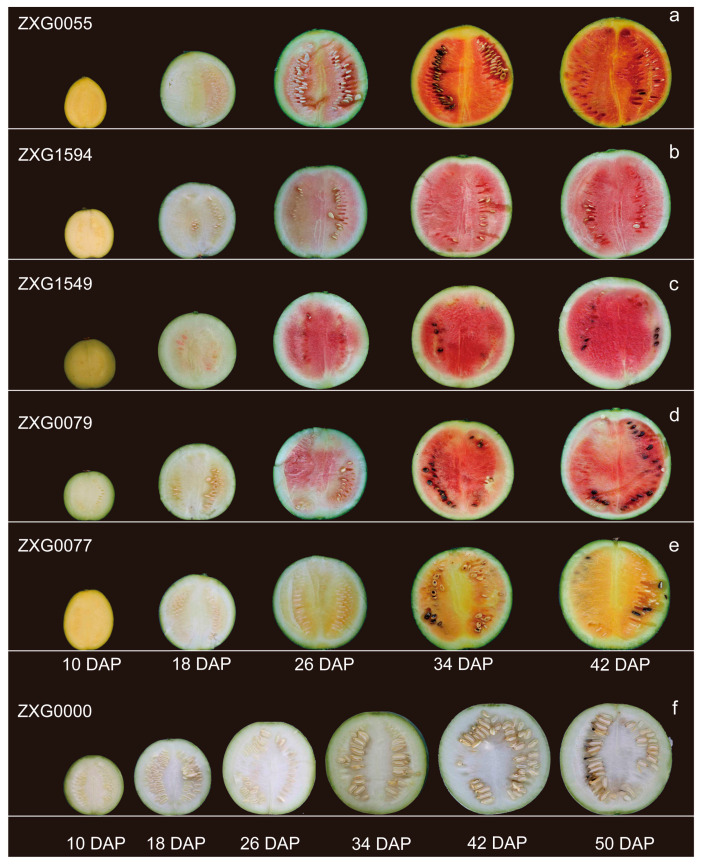
Photographs of six watermelon accessions with different flesh colors at five or six time points (10, 18, 26, 34 and 42 or 50 days after pollination) during fruit development. (**a**–**f**): ‘ZXG0055’, ‘ZXG1594’, ‘ZXG1549’, ‘ZXG0079’, ‘ZXG0077’and ‘ZXG0000’, respectively.

**Figure 3 genes-13-01189-f003:**
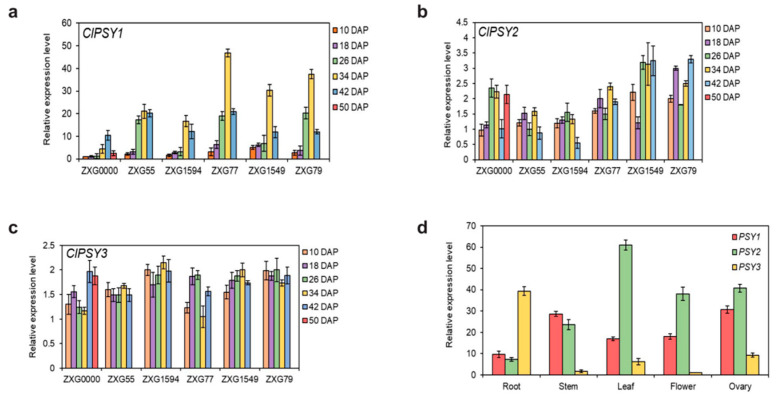
Expression of *ClPSY* members in different watermelon accessions and organs. (**a**–**c**) Transcription levels of *ClPSY1*, *ClPSY2* and *ClPSY3* in six watermelon varieties with different fruit colors at six time points (10, 18, 26, 34, 42, 50 days after pollination) during fruit development. Additionally, the fruit sample ‘ZXG0000’ at 10 DAP was used for calibration. The bars represent the means ± SD (*n* = 3). (**d**) Expression patterns of the three *ClPSY* genes in different organs. Total RNA samples were isolated from various watermelon organs, including 4-week seedling roots, stems, leaves, flowers and ovaries. The y-axis shows the relative expression levels of the three *ClPSY* genes in different tissues compared to *ClPSY3* in flowers. The bars represent the means ± SD (*n* = 3).

**Figure 4 genes-13-01189-f004:**
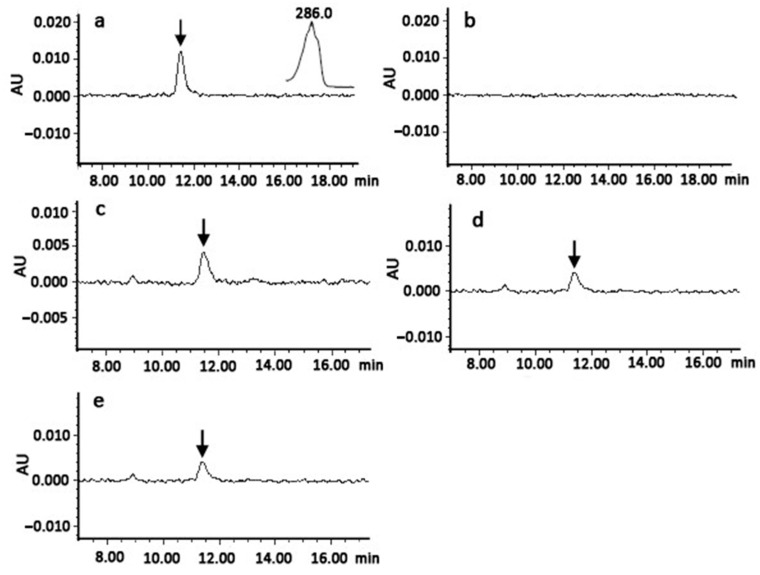
Functional complementation of watermelon ClPSY proteins. (**a**) *E. coli* cells harboring EB were used as a positive control. The peak represents phytoene (indicated by an arrow); (**b**) E+pET (empty vector) was used as a negative control; (**c**) E+ pET-*ClPSY1*; (**d**) E+ pET-*ClPSY2*; (**e**) E+ pET-*ClPSY3*. The peak representing phytoene (indicated by an arrow) was observed in cells expressing *ClPSY1*, *ClPSY2* and *ClPSY3*.

**Figure 5 genes-13-01189-f005:**
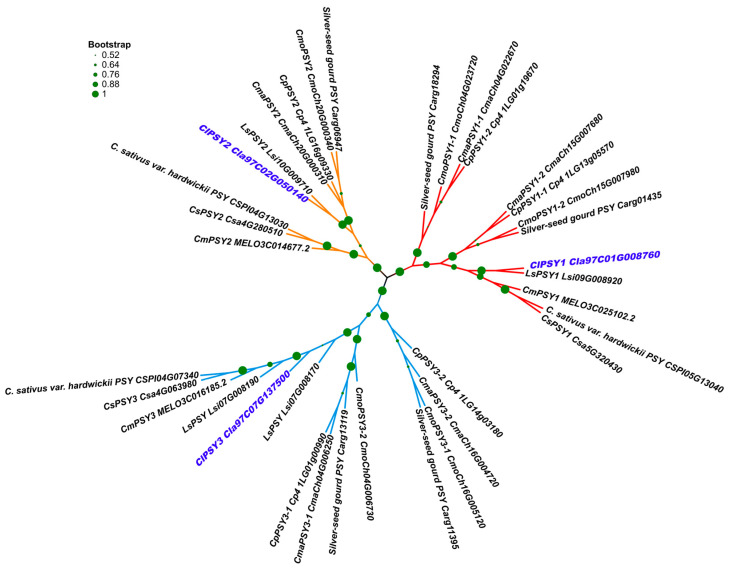
Phylogenetic tree and gene structure of *PSY* family proteins from various species retrieved from the Cucurbit Genomics Database (http://www.icugi.org/; accessed on 15 January 2022).

**Figure 6 genes-13-01189-f006:**
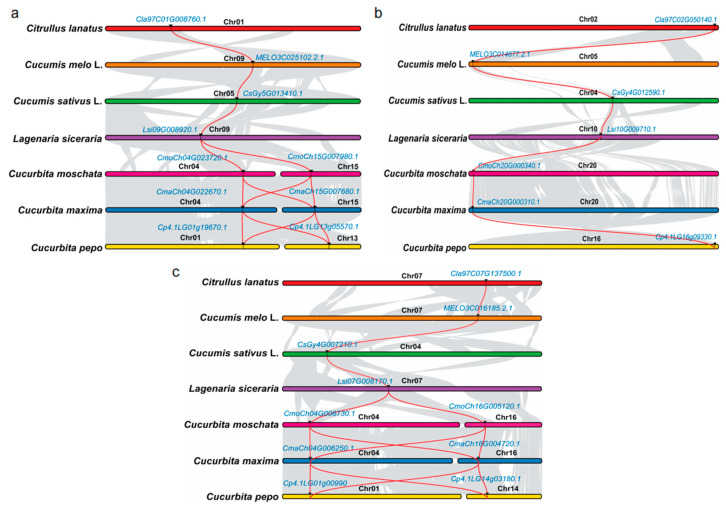
The synteny analysis of PSY members among different cucurbit crops. (**a**–**c**) for the *ClPSY1*, *ClPSY2* and *ClPSY3* homologous genes in different cucurbit crops.

**Figure 7 genes-13-01189-f007:**
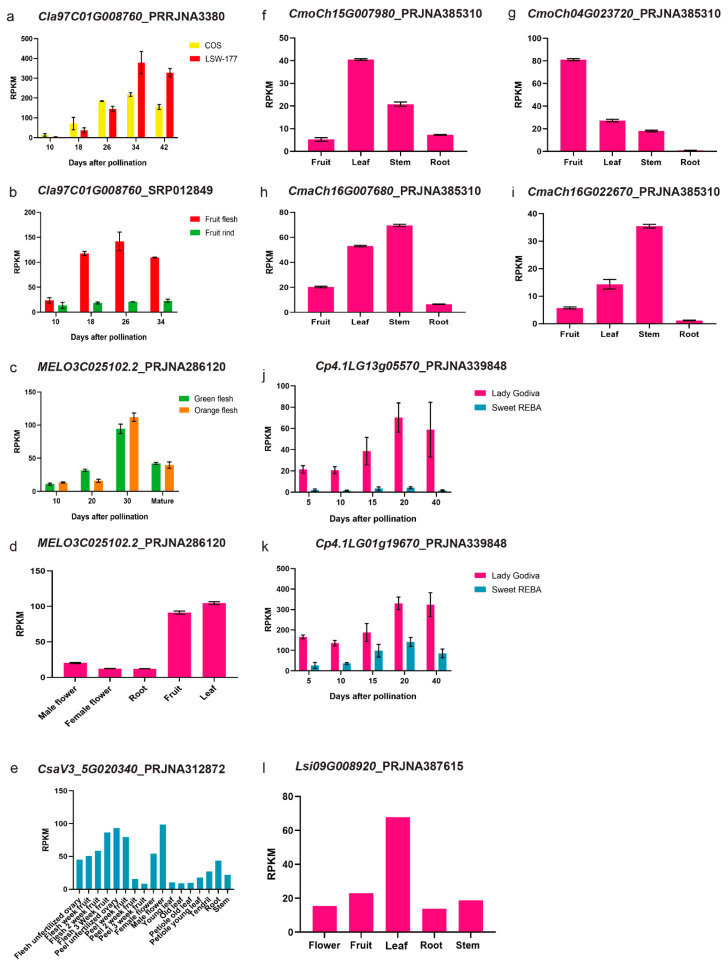
*ClPSY1* homologous genes expression pattern analysis with the published RNA-seq data. (**a**–**l**): RPKM of PRRJNA3380, SRP012849, PRJNA286120, PRJNA312872, PRJNA385310, PRJNA339848 and PRJNA387615.

**Figure 8 genes-13-01189-f008:**
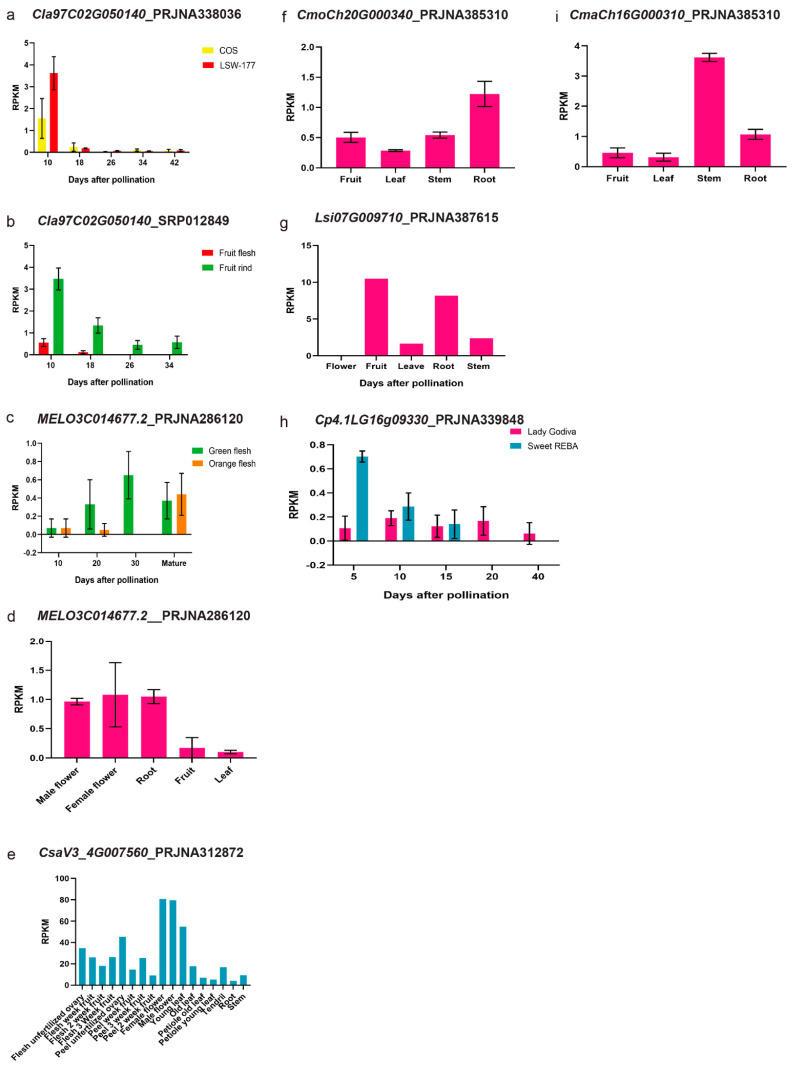
*ClPSY2* homologous genes expression pattern analysis with the published RNA-seq data. (**a**–**i**): RPKM of PRJNA338036, SRP012849, PRJNA286120, PRJNA312872, PRJNA385310, PRJNA387615 and PRJNA339848.

**Figure 9 genes-13-01189-f009:**
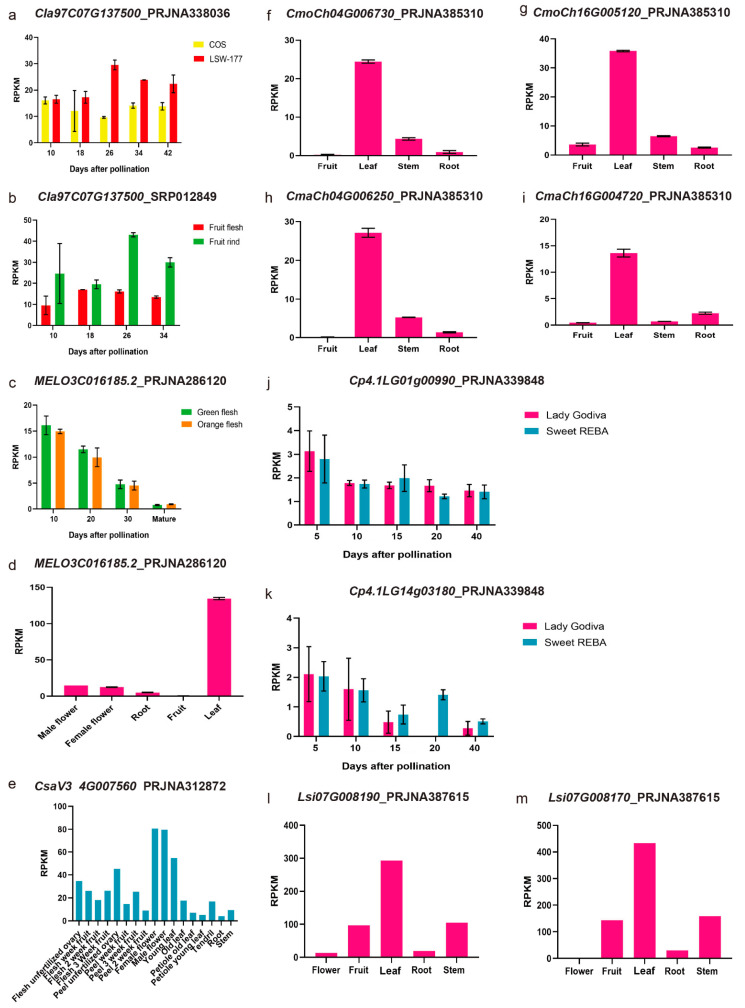
*ClPSY3* homologous genes expression pattern analysis with the published RNA-seq data. (**a**–**m**): RPKM of PRJNA338036, SPR012849, PRJNA286120, PRJNA312872, PRJNA385310, PRJNA339848 and PRJNA387615.

**Table 1 genes-13-01189-t001:** Description of watermelon *PSY* family genes and *ClYLS8*.

Gene	Gene ID	Chromosome	Positions
*ClPSY1*	*Cla97C01G008760*	1	Chr01: 9448426 ~ 9451311 (+)
*ClPSY2*	*Cla97C02G050140*	7	Chr07: 25039090 ~ 25042269 (+)
*ClPSY3*	*Cla97C07G137500*	2	Chr02: 37419674 ~ 37421665 (−)
*ClYLS8*	*Cla97C02G038590*	2	Chr02: 22823405 ~ 22824590 (−)

**Table 2 genes-13-01189-t002:** Primers for amplifying the full-length cDNA and primers used for qRT-PCR analysis.

Primer Name	Amplicon Size (bp)	Primer Sequence (5′-3′)
ClPSY1	1266	F: GGAATTCATGTCTTTTGCTCCTTCGTTGG R: CCTCGAGAATTCATGAAGGGCCAAG
ClPSY2	1194	F: GGAATTCATGTCTGGTGTGAATGCCAACTCTC R: CCTCGAGCTATCTTGTTACCAAATCTTT
ClPSY3	1068	F: CGGGATCCATGAGCAAAAGTGGGGTAATT R: CCTCGAGTCAATGGAAGACTAGACTGGGTGCC
qPSY1	179	F: TAAGTTTCCAGTTGATATTCAGCCG R: GTGCTTGCTTGGGAGTCAGG
qPSY2	132	F: CCTGTCATGGGATTGGCACC R: CTCTTCCCCTCCTAGCATCTTCTCC
qPSY3	118	F: GCACCTGATTCTTCACTTCCTACTC R: ATCCTACCCCTTATAGCATCCTCTC
ClYLS8	74	F: AGAACGGCTTGTGGTCATTC R: GAGGCCAACACTTCATCCAT

**Table 3 genes-13-01189-t003:** *PSYs* in different *Cucurbit* family crops.

Species	Gene ID	Position
*Cucumis melo*	*MELO3C025102.2*	Chr09: 14553133 ~ 14556862 (+)
	*MELO3C014677.2*	Chr05: 487434 ~ 489932 (+)
	*MELO3C016185.2*	Chr07: 20315779 ~ 20319205 (+)
*Cucumis sativus*	*CsaV3_5G020340*	Chr05: 15246891 ~ 15249568 (+)
	*CsaV3_4G023380*	Chr04: 13627834 ~ 13631557 (+)
	*CsaV3_4G007560*	Chr04: 5185258 ~ 5189102 (−)
*Cucumis sativus* var. *hardwickii*	*CSPI05G13040*	Chr05: 12570152 ~ 12574113 (+)
	*CSPI04G07340*	Chr04: 5188671 ~ 5192373 (−)
	*CSPI04G13030*	Chr04: 11240846 ~ 11244106 (+)
*Lagenaria siceraria*	*Lsi07G008190*	Chr07: 9123302 ~ 9126554 (−)
	*Lsi09G008920*	Chr09: 10525942 ~ 10528939 (+)
	*Lsi07G008170*	Chr07: 9114307 ~ 9117400 (−)
	*Lsi10G009710*	Chr10: 14062470 ~ 14067420 (+)
*Cucurbita maxima* var. *Rimu*	*CmaCh16G004720*	Chr16: 2390509 ~ 2394475 (+)
	*CmaCh15G007680*	Chr15: 3777993 ~ 3782451 (−)
	*CmaCh04G022670*	Chr04: 15833307 ~ 15835991 (+)
	*CmaCh04G006250*	Chr04: 3203989 ~ 3206323 (−)
	*CmaCh20G000310*	Chr20: 140674 ~ 142255 (+)
*Cucurbita moschata* var. *Rifu*	*CmoCh04G023720*	Chr04: 17691476 ~ 17694558 (+)
	*CmoCh16G005120*	Chr16: 2472841 ~ 2477006 (+)
	*CmoCh15G007980*	Chr15: 3929562 ~ 3934079 (−)
	*CmoCh04G006730*	Chr04: 3346882 ~ 3349011 (−)
	*CmoCh20G000340*	Chr20: 224511 ~ 225982 (+)
*Cucurbita pepo* subsp. *pepo*	*Cp4.1LG14g03180*	Chr14: 2471728 ~ 2475409 (+)
	*Cp4.1LG13g05570*	Chr13: 5416613 ~ 5421307 (+)
	*Cp4.1LG01g19670*	Chr01: 16838967 ~ 16841930 (+)
	*Cp4.1LG01g00990*	Chr01: 3271567 ~ 3274741 (−)
	*Cp4.1LG16g09330*	Chr16: 8547559 ~ 8548969 (−)
*Cucurbita argyrosperma*	*Carg18294*	Scaffold_120: 491768 ~ 494735 (−)
	*Carg06947*	Scaffold_065: 466719 ~ 468129 (−)
	*Carg13119*	Scaffold_101: 425459 ~ 427786 (+)
	*Carg01435*	Scaffold_004: 945560 ~ 949123 (+)
	*Carg11395*	Scaffold_025: 739796 ~ 743637 (+)

## Data Availability

Not applicable.

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
