# Peer review of "Identification and Characterization Roles of Phytoene Synthase (PSY) Genes in Watermelon Development"

_genes, 2022, doi:10.3390/genes13071189_

Round 1
Reviewer 1 Report
Although this manuscript provides valuable result, its format needs revised.
1. Page 1, line2: role should be capitalized.
2. Page 1, lines 5-8: Please provide the e-mails of the first and second authors.
3. Page 1, line 9: Typo: aythor’s.
4. Page 1, line 31: Reference format (Yuan et al., 2015) is incorrect, please use numeral system. Also correct reference format in all manuscript.
5. Page 1, line 45: ‘However’ should be lower case.
6. Page 3, line 98: Place 2.1 in front of Plant materials. Line 110 and line 120: place 2.2 and 2.3, respectively. Also correct format in all manuscript.
7. Page 6, line 276: E. coli cells harboring EB were…. It is a little bit confused. As I read in the main text (line 266), EB is an E. coli strain, right?
8. Page 10, Figure 7.
Subfigure a: mild suggestion, the authors already used days after pollination as the title of x-axis. Numbers only (10, 18, …) may be simplified and clear.
Subfigure b: typos, 10DPA, 18DPA,…etc.
Subfigure c: typos, 10DPA, 20DPA,…etc.
Subfigures e and l: Standard deviation is unseen, why? Is sample size one? In subfigure l, the authors use ‘leave’ instead of ‘leaf’ in subfigures d, f, h, g and i.
9. Figures 8 and 9 have the same issues as Figure 7. Please carefully revise them.
Reviewer 2 Report
This is an interesting manuscript about the role of phytoene synthase (PSY) in the biosynthesis of carotenoid and flesh color formation in watermelon with the identification of three ClPSY genes through the watermelon genome and characterization of the complete protein-coding sequences (CDS). Also, a heterologous omplementation system was used to study the ClPSY gene family with regard to enzyme activity. The manuscript shows novel results, and the objectives are very clear. The results are well presented, and their interpretation is relevant.
Author Response
Thank you very much for your recognition of our manuscript and work. Thanks.

Round 2
Reviewer 1 Report
The authors addressed all my comments. The manuscript is acceptable!